# Lightweight One-Stage Maize Leaf Disease Detection Model with Knowledge Distillation

**Yanxin Hu** [1]**, Gang Liu** [1,2]**, Zhiyu Chen** [1,2]**, Jiaqi Liu** [1] **and Jianwei Guo** [1,*]

1   School of Computer Science and Engineering, Changchun University of Technology, Changchun 130102, China
2   Jilin Province Data Service Industry Public Technology Research Centre, Changchun 130102, China
*   Correspondence: guojianwei@ccut.edu.cn

**Abstract:** Maize is one of the world's most important crops, and maize leaf diseases can have a direct impact on maize yields. Although deep learning-based detection methods have been applied to maize leaf disease detection, it is difficult to guarantee detection accuracy when using a lightweight detection model. Considering the above problems, we propose a lightweight detection algorithm based on improved YOLOv5s. First, the Faster-C3 module is proposed to replace the original CSP module in YOLOv5s, to significantly reduce the number of parameters in the feature extraction process. Second, CoordConv and improved CARAFE are introduced into the neck network, to improve the refinement of location information during feature fusion and to refine richer semantic information in the downsampling process. Finally, the channel-wise knowledge distillation method is used in model training to improve the detection accuracy without increasing the number of model parameters. In a maize leaf disease detection dataset (containing five leaf diseases and a total of 12,957 images), our proposed algorithm had 15.5% less parameters than YOLOv5s, while the mAP(0.5) and mAP(0.5:0.95) were 3.8% and 1.5% higher, respectively. The experiments demonstrated the effectiveness of the method proposed in this study and provided theoretical and technical support for the automated detection of maize leaf diseases.

**Keywords:** maize disease detection; knowledge distillation; object detection; YOLOv5s

## 1. Introduction

Maize, grown over a land area that ranks second only to rice and wheat, serves as a significant food source for humans, as well as a crucial feed for livestock. Moreover, it holds substantial value as a raw material in the light and medical industries [1]. The primary factor contributing to the reduction in corn yields is leaf disease. Hence, the expeditious and precise identification of these ailments, facilitating early detection, enables cultivators, breeders, and researchers to efficiently employ suitable preventive measures, in order to alleviate the impact of the disease. The acquisition of the specialized knowledge necessary for the identification of diseases in maize leaves poses a significant challenge for the average cultivator. This often leads to erroneous diagnoses of maize plant diseases, which can have substantial negative impacts on the economy. The advancement of machine vision and deep learning has facilitated the automated identification and diagnosis of plant pests and diseases [2].

In recent years, there has been a notable emphasis among scholars on the application of machine learning and image processing techniques for plant disease identification. Support vector machines (SVM) were proposed by Cortes and Vapnik [3] in 1995, and Yang et al. [4] multiple begetter SVM (MBSVM) further improved the classification ability. Bhange and Hingoliwala [5] applied an SVM model to identify pomegranate leaf disease, by extracting features based on parameters such as color, morphology, and CCV, and through clustering using a k-means algorithm. Thomas et al. [6] used an SVM classifier to classify and identify

hyperspectral images of potato late blight. This K-nearest neighbor (KNN) algorithm was proposed by [7] in 1975. Similarly to SVM, this method can be applied to multiple classes. Ref. [8] used this model to classify and identify deadly fungal (Ganoderma) diseases in oil palm plantations. Zhang et al. [9] segmented maize leaf patches and extracted disease feature vectors, and used KNN to identify five different diseases of maize leaves. Devi et al. [10] used a HOG (histogram of oriented gradients) and KNN classifier for accurate detection and classification of peanut foliar diseases. The conventional algorithms utilized in machine vision for the automated diagnosis of plant leaf diseases are characterized by their inherent challenges, including complexity, susceptibility to errors, and reliance on manual feature extraction. Consequently, these limitations hinder the effectiveness and practicality of disease-detection processes.

It was observed by researchers that convolutional neural networks (CNNs) exhibited the ability to acquire feature representations that possess both robustness and expressiveness. This realization came about when CNN-based classification networks demonstrated exceptional performance in the 2012 ILSVRC image classification competition [11]. Consequently, certain scholars have employed deep learning techniques in the context of identifying and detecting diseases in crop leaves [2,12]. Richey et al. [13] proposed a model based on ResNet50 for the detection of northern maize leaf blight in maize plants. The model was applied in a mobile phone application. Zhang et al. [14] used an improved GoogleNet to identify diseases in the leaves of maize plants. Wu et al. [15] achieved better results by capturing images of maize planting sites using UAVs and using convolutional neural networks for classification. Panigrahi et al. [16] presented improved CNN models for training and testing four maize leaf images, by adding modified linear unit activation functions and Adam optimizers. Despite having good accuracy, CNN-based network models are not very efficient. This has motivated several researchers to concentrate on finding ways to improve CNN-based network models. Howard et al. [17] produced MobileNetV1, which uses deep separable convolution and is a lightweight CNN. Since then, MobileNetV2 [18] and MobileNetV3 [19] have been proposed, to improve the effectiveness of MobileNet. In addition, there are other lightweight networks with better performance, such as ShuffleNet [20], ShuffleNetV2 [21], and GhostNet [22], proposed by Huawei's Noah's Ark Lab. By introducing an ECA [23] attention mechanism and using cross-entropy loss, Bi et al. [24] proposed an improved MoblieNet network to identify maize leaf diseases, with good results. In their study, Gulzar [25] introduced a classification method that utilizes an enhanced version of the MobileNetV2 architecture for fruit recognition. In their study, Dhiman et al. [26] thoroughly examined the techniques employed in image capture, preprocessing, and classification for the detection of citrus fruit diseases. These techniques encompassed both machine learning and deep learning methodologies. The program demonstrated a remarkable accuracy rate of 99%. Chen et al. [27] proposed an improved ShufflenetV2 network to identify apple leaf diseases, which is lighter but loses accuracy. In a separate domain of research, Aggarwal et al. [28] used integration methodologies to enhance the efficacy of the prevailing convolutional neural networks. They further employed a pre-existing model to suggest a deep learning framework centered on layered integration, thus yielding classifiers that are both more dependable and robust. In practical applications, the utilization of image classification for the purpose of identifying leaf disease in crops has yielded a relatively lower amount of information. Conversely, object detection techniques have the capability of precisely determining the location of the lesion, but these pose greater challenges compared to image classification methods.

Girshick et al. [29] proposed an R-CNN model with CNN features, to facilitate a combination of deep learning and object detection. The Fast R-CNN [30] aggregates a region of interest (RoI) layer to improve speed and accuracy but still uses external algorithms to extract object candidate frames. The Faster R-CNN [31] model proposes a region proposal network (RPN) that discards the manual extraction of candidate frames and merges the object candidate frame extraction process into a single deep network. Du et al. [32] proposed an improved Faster R-CNN for maize pest detection. Kumar et al. [33] proposed a

Faster R-CNN approach to identify 93 maize diseases. He et al. [34] used MFaster R-CNN for maize leaf disease detection. Since this type of candidate area-based detector requires two steps, it is collectively referred to as a two-stage object detection algorithm. This type of algorithm has good detection accuracy but lacks real-time performance. A one-stage object detection algorithm is a good solution to the problem of a lack of real-time detection compared to the two-stage, especially with the YOLO (You Only Look Once) [35–43] series, which is set to become the mainstream of object detection.Shill and Rahman [44] used YOLOv3 and YOLOv4 as expert systems to detect plant diseases. Mamat et al. [45] employed the enhanced YOLOv5 algorithm to accurately detect a set of 100 photos depicting the olea europaea palm. The achieved mean average precision (mAP) for this detection task was 98.7%. Liu et al. [46] proposed a YOLOX-based algorithm for tomato leaf disease detection, which uses MobileNetV3 instead of the YOLOX backbone for lightweight model feature extraction and introduces a CBAM [47] attention mechanism. While real-time operation is achievable with one-stage object detection algorithms, their implementation on mobile and edge devices remains a challenging task. Various researchers have proposed alternative approaches in the form of lightweight object detection methods, to address this concern. Examples include YOLOv3-tiny [48], YOLOv4-tiny [49], and YOLOv5-lite [50]. Li et al. [51] proposed a YOLOv4-tiny based algorithm for broken corn kernel detection.

The YOLO family has been extensively employed across various domains, including agriculture (for the purpose of diagnosing crop pests and illnesses), industry, and medical. The YOLOv5s model, known for its compact size, is a suitable choice for environments with restricted computational resources. Despite its simplified model size and computational complexity compared to other lightweight detection networks, YOLOv5s demonstrates commendable accuracy and robustness on various target detection benchmark datasets. This compelling performance motivated us to employ YOLOv5s as a benchmark model for identifying maize leaf diseases. In order to further advance our research, we considered the following knowledge gaps:

1. The predominant focus of contemporary research pertaining to the identification of maize leaf diseases lies in the domain of image classification recognition. However, it is important to note that this approach offers a restricted amount of information. Alternatively, employing object detection techniques can yield a more comprehensive set of data, hence facilitating more effective disease treatment strategies;

2. The absence of well-defined evaluation measures or benchmarks especially designed for the identification of maize leaf diseases poses challenges in assessing the performance of various object detection algorithms for this purpose;

3. There is a limited number of maize leaf disease detection models that simultaneously address detection accuracy and speed, hence hindering the practical implementation of these methods.

Li et al. [52] presented a algorithm for the identification of vegetable diseases in their research. The researchers employed the YOLOv5s architecture as a foundational framework but introduced modifications by substituting the CSP module with cross stage partials integrated with a transformer encoder. Furthermore, enhancements were made to the inception module within the neck network. The algorithm under consideration demonstrated a mean average accuracy (mAP) of 0.5 on the curated vegetable dataset, thereby surpassing the performance of the original YOLOv5s by a margin of 4.8%. Nevertheless, it is important to acknowledge that the algorithm that underwent modifications led to a decrease of 1.3 megabytes in the parameter count, thus affecting the frames per second (FPS) measure. In their study, Zhang et al. [53] employed the YOLOv5-lite model to accurately detect tea leaves suitable for harvesting by an automated picking robot. The YOLOv5-lite model incorporates an enhanced shufflenet network, as a substitute for the backbone network utilized in the original YOLOv5 model. This substitution results in a notable enhancement in detection speed; however, this comes at the expense of a considerable reduction in detection accuracy. Run-Hua et al. [54] utilized an increased attention mechanism and ghost convolution techniques in their research to improve the performance

of the YOLOv5s network. This led to a significant increase of 2.6% in the mean average precision (mAP) on the VOC dataset, in comparison to the initial YOLOv5s network. It is noteworthy to mention that the observed enhancement in performance was attained with an augmentation of 0.75 M in the size of the model file. Based on the previously mentioned research, it becomes evident that incorporating a lightweight module as a replacement for the module in the original YOLOv5s network results in a certain reduction in accuracy. Conversely, the inclusion of a module aimed at enhancing accuracy entails a compromise in terms of detection speed. One notable distinction between this study and the afore-mentioned enhancements lies in the substantial reduction in model parameters achieved through the incorporation of an enhanced lightweight convolutional module. While this reduction results in a certain degree of diminished detection accuracy, the integration of an improved lightweight upsampling algorithm and CoordConv within the neck network serves to augment the network's perceptual capabilities and enhance its sensitivity to the detected location. Consequently, this mitigates the aforementioned loss in detection accuracy. Furthermore, the channel-wise knowledge distillation approach enhances the detection accuracy of the network, without an increase in the number of model parameters.

In conjunction with the aforementioned studies, it is evident that the current algorithms for detecting maize leaf diseases fail to strike a balance between accuracy and speed of detection. The primary objective of this study was to maximize the level of accuracy within a comparatively lightweight detection algorithm, hence enhancing the applicability of the maize leaf disease detection method over a wider range of settings. In order to accomplish this objective, this study employs a lightweight algorithm for detecting maize leaf diseases. The program achieves enhanced detection accuracy by incorporating lighter modules, enhancing the upsampling technique, and employing knowledge distillation. Initially, the YOLOv5s model underwent a modification wherein the CSP module was substituted with a more lightweight variant known as Faster-C3. This substitution resulted in a reduction in model parameters and computational complexity by minimizing the number of convolution calculations. Consequently, a slight compromise in accuracy was incurred. Furthermore, the neck architecture employed in YOLOv5s incorporates the Co-ordConv and an enhanced lightweight CARAFE downsampling module. This integration serves to enhance the semantic information within the feature fusion process, resulting in improved detection accuracy, while maintaining minimal alterations to the model parameters. The proposed algorithm incorporates knowledge from YOLOv5m through a channel-wise knowledge distillation method during training, resulting in an enhancement of the detection accuracy, without causing any damage to the model parameters. The main contributions and innovations of this work are summarized below:

(1) In this study, we propose an enhanced and more efficient lightweight detection algorithm that builds upon the YOLOv5s framework. Our approach involves replacing the original CSP module with the Faster-C3 module, resulting in a significant reduction in the total number of model parameters. The proposed enhancements in the neck network involve the integration of CARAFE and CoordConv modules. These modifications aim to enhance the extraction of semantic information, with a particular focus on improving the accuracy of detecting object locations;

(2) Additional enhancements to the initial model are achieved by implementing the channel-wise knowledge distillation technique during training and adopting the WIoU metric as a loss function. These modifications effectively enhance the accuracy of maize leaf disease detection, without introducing any additional model parameters or computational complexity;

(3) A series of comparative experiments were carried out on a dataset pertaining to the detection of maize leaf disease. The findings of these experiments indicate that our proposed algorithm exhibits superior comprehensive performance when compared to other lightweight algorithms. The algorithm has the capability to offer precise planting, visual management, and intelligent decision-making for maize crops.

## 2. Materials and Methods

### 2.1. Data Source

The data sources for this study included maize disease image data obtained from PlantVillage [55]. The dataset comprises five prevalent diseases affecting maize leaves, specifically Cercospora zeaemaydis Tehon and Daniels, Puccinia polysora, Common Rust, Blight, and Mycosphaerella maydis Lindau. The dataset underwent annotation in the PascalVOC format, and subsequent augmentation was performed on the original dataset utilizing image processing techniques, including Gaussian noise addition and image rotation. Table 1 shows the details of the dataset. Corn leaf disease bounding boxes are used to identify and locate infected corn leaf areas in an image. A bounding box can be used to detect and segment leaf disease areas for further disease analysis and processing. Using pixel coordinates to represent the bounding box, a bounding box can be represented using four values (x_min, y_min, x_max, y_max), where (x_min, y_min) are the coordinates of the upper left corner of the rectangular bounding box and (x_max, y_max) are the coordinates of the lower right corner of the rectangular bounding box. Figure 1 visually represents five examples of maize leaf diseases. The final dataset comprised a total of 12,957 images, which were allocated in a ratio of 7:1:2 among training, validation, and test sets.

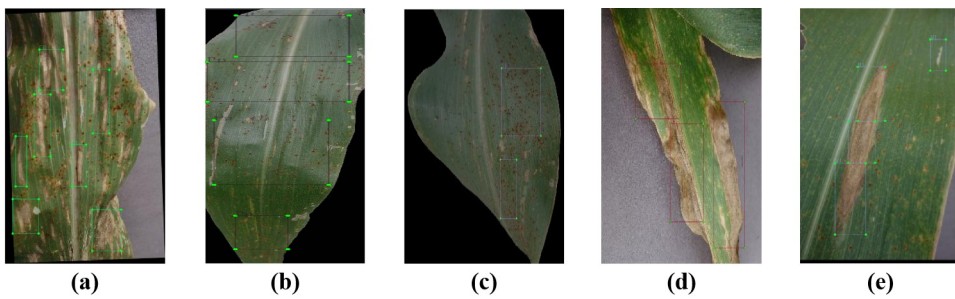

| **(a)** | **(b)** | **(c)** | **(d)** | **(e)** |

**Figure 1.** Pictures with labels of the five diseases of corn leaves. (**a**): Cercospora zeaemaydis Tehon and Daniels; (**b**): Puccinia polysora; (**c**): Common Rust; (**d**): Blight; (**e**): Mycosphaerella maydis Lindau.

**Table 1.** Information table about the quantity of the different types of maize leaf disease picture data.

| Name | Number of Original Images | Number of Enhanced Images |
|---|---|---|
| Cercospora zeaemaydis Tehon and Daniels | 462 | 1386 |
| Puccinia polysora | 831 | 2493 |
| Common Rust | 1179 | 3537 |
| Blight | 1145 | 3435 |
| Mycosphaerella maydis Lindau | 702 | 2106 |

In this study, PascalVOC, a common public dataset for target detection, was used, containing a total of 21,504 images from PascalVOC 2012 and PascalVOC 2007, covering 20 different object categories, such as people, cars, planes, cats, dogs, and so on. Each image is labeled with the object category and a bounding box. It was divided according to a training set, validation set, and test set ratio of 7:1:2.

### 2.2. The Basic YOLOv5

The network architecture of YOLOv5 is depicted in Figure 2, comprising three primary components: the backbone network, the neck network, and the detection head. The backbone network of YOLOv5 draws inspiration from CSPDarkNet [56] and primarily serves the purpose of extracting image features. The neck network comprises of PANet [57] and FPN [58], which has been specifically devised to effectively leverage both high and low semantic information during the process of feature fusion. The ultimate outcome of the detection process is derived from the output generated by the convolutional layer.

The YOLOv5 model employed a combined loss function to facilitate training of the bounding box regression, classification, and confidence aspects. The IoU-Loss was employed as a loss function for quantifying the similarity between the predicted bounding box and the actual annotation. This is considered the most commonly utilized metric for assessing the similarity between bounding boxes. Several IoU calculations have been proposed by various researchers, including DIoU [59], CIoU [59], GIoU [60], and WIoU [61]. In light of the practical considerations surrounding the detection of maize leaf disease, we proposed the utilization of the weighted intersection over union (WIoU) metric as an alternative to the original complete intersection over union (CIOU) metric. This decision was based on the fact that WIoU employs a dynamic non-monotonic focusing mechanism, coupled with a gradient gain allocation strategy. This approach effectively mitigates the adverse effects of anchor frames of subpar quality, enabling WIoU to prioritize the detection of such frames and ultimately enhance the overall performance of the detector.

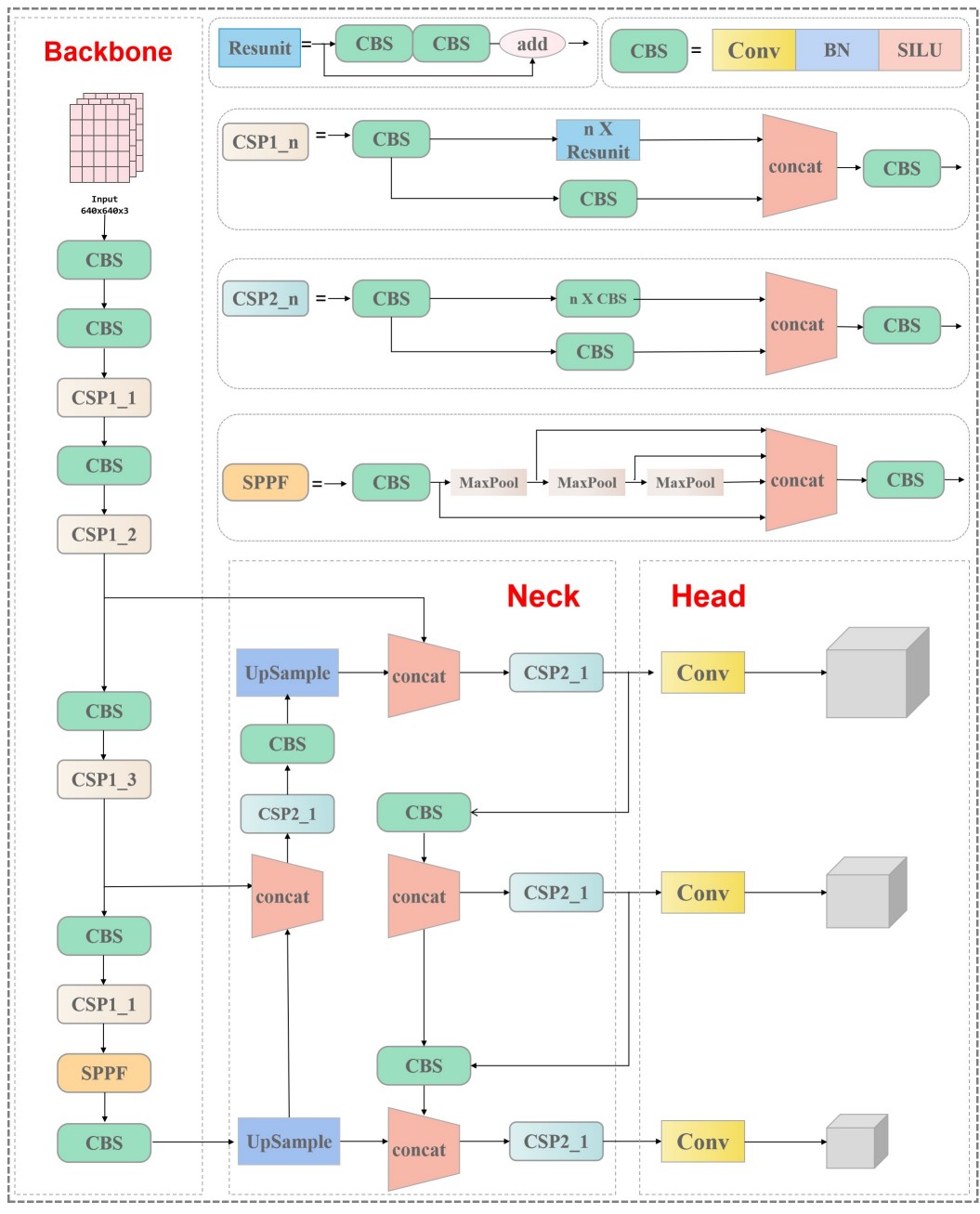

**Figure 2.** YOLOv5 network structure.

There are five iterations of YOLOv5, which are referred to as YOLOv5n, YOLOv5s, YOLOv5m, YOLOv5l, and YOLOv5x. All of the previously mentioned models have the same network structure, although they differ in terms of their network depth and feature map width. This feature allows scalability of the network model, thereby facilitating the ability to adjust the number of parameters within the network, in order to regulate the size of the model. The performances corresponding to their particular cases are displayed in Table 2. The YOLOv5n model is renowned for its efficient design, prioritizing computational efficiency. However, it is crucial to acknowledge that this model also demonstrates the least accurate detection performance. The YOLOv5x model demonstrates a superior detection accuracy compared to the other models, albeit at the cost of a relatively slower detection speed. When evaluating the detection of maize leaf disease, the YOLOv5s model is identified as the most suitable detection network, owing to its efficient design and capacity to uphold a satisfactory level of detection precision.

**Table 2.** Performance comparison of the different models of YOLOv5 on the COCO dataset.

| Model | mAP(0.5) | mAP(0.5:0.95) | Speed v100 (ms) | Params (M) | FLOPs (G) |
|---|---|---|---|---|---|
| YOLOv5n | 45.7 | 28.0 | 0.6 | 1.9 | 4.5 |
| YOLOv5s | 56.8 | 37.4 | 0.9 | 7.2 | 16.5 |
| YOLOv5m | 64.1 | 45.4 | 1.7 | 21.2 | 49.0 |
| YOLOv5l | 67.3 | 49.0 | 2.7 | 46.5 | 109.1 |
| YOLOv5x | 68.9 | 50.7 | 4.8 | 86.7 | 205.7 |

### 2.3. Improved YOLOv5

2.3.1. Faster-C3 Module

The initial implementation of the CSP module in YOLOv5 necessitated the use of multiple convolution operations, leading to a notable augmentation in the parameter number and computational complexity. The rationale behind conducting this study was derived from the FasterNet [62] model, which served as the impetus for suggesting the Faster-C3 module as a potential substitute for the CSP module. The PConv architecture represents an enhanced convolutional framework that facilitates the selective extraction of spatial features from a specific subset of input channels, while simultaneously preserving the remaining channels. This stands in contrast to conventional convolutional techniques. When dealing with situations that require sequential or conventional memory access, it is common practice to select either the first or last consecutive channel as a representative of the entire feature mapping. In this particular scenario, the computation solely considers inputs and outputs that possess an equal number of channels. In order to enhance the utilization of information from various channels, a pointwise state convolution (PWConv) is incorporated into PConv. A T-shaped convolution pattern is observed in the effective receptive domain of the input feature map, as depicted in Figure 3b. According to the illustration in Figure 3c, it is evident that this particular pattern places a higher degree of importance on the central position, in contrast to the traditional convolution method.

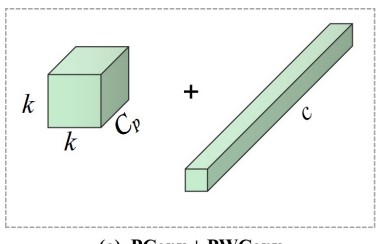 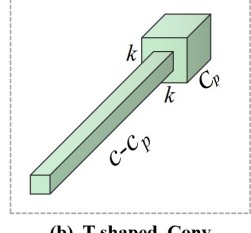 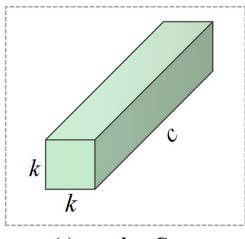

(a) PConv + PWConv   (b) T-shaped Conv   (c) regular Conv

**Figure 3.** PConv and PWConv structures.

Each Faster-C3 model is equipped with a PConv layer, which is subsequently followed by two convolutional layers with a kernel size of $1 \times 1$. The combined entities are presented

as inverted residual blocks, wherein the central layer possesses a greater number of channels and a shortcut connection is established to recycle the input features. A shortcut connection is implemented to facilitate the reuse of input features. The Faster-C3 module is depicted in Figure 4.

The left side of Figure 4 provides an explanation of the operating principle of PConv. The PConv algorithm conducts spatial feature extraction by selectively applying normal convolutional operations to a subset of the input channels, while preserving the remaining channels. In the case of consecutive or regular memory accesses, it is common practice to select either the first or the last consecutive channel to serve as a representation of the whole feature mapping. In order to ensure wide applicability, it is assumed that the mappings of input and output features possess an equal number of channels. Therefore, the floating point operations (FLOPs) associated with a pconv are limited to

$$h \times w \times k^2 \times c_p^2. \tag{1}$$

With a typical partial ratio $r = c_p/c = 1/4$, the FLOPs of a PConv are only 1/16 of a regular Conv. In addition, PConv has a smaller amount of memory access, i.e.,

$$h \times w \times 2c_p + k^2 c_p^2 \approx h \times w \times 2c_p. \tag{2}$$

which is only 1/4 of a regular Conv for $r = 1/4$.

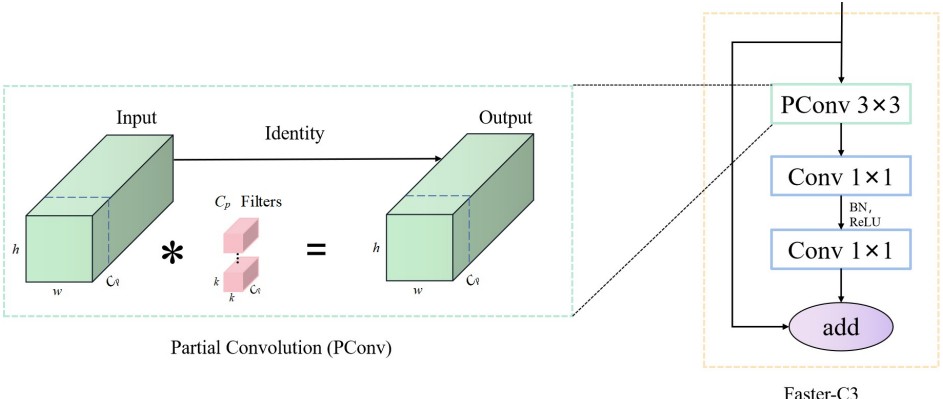

**Figure 4.** Faster-C3 module.

### 2.3.2. Improved Neck Network

In the context of maize leaf disease detection, accurately determining the precise location of the disease is a crucial objective in the detection process. On the other hand, traditional convolutional methods lack the capability to transform the spatial representation into Cartesian coordinates and one-hot pixel coordinates. Convolution exhibits equivariance, which implies that it lacks awareness of the specific spatial location of each filter during its application to the input. The CoordConv method [63] is introduced as a means to address the issue of positional information in feature extraction, without introducing additional parameters in comparison to standard convolution. The CoordConv technique can enhance the process of convolution by providing spatial coordinates to inform the network about the precise locations of the filters. The aforementioned procedure is accomplished by incorporating two additional channels into the input, one corresponding to the i-coordinate and the other to the j-coordinate. The convolution's sensitivity to position is enhanced through the incorporation of coordinate addition. In this research, CoordConv (as depicted in Figure 5) was employed to substitute the conventional convolution in the original YOLOv5s neck network. This substitution aimed to improve the extraction of location information related to maize diseases, consequently leading to an enhancement in detection accuracy.

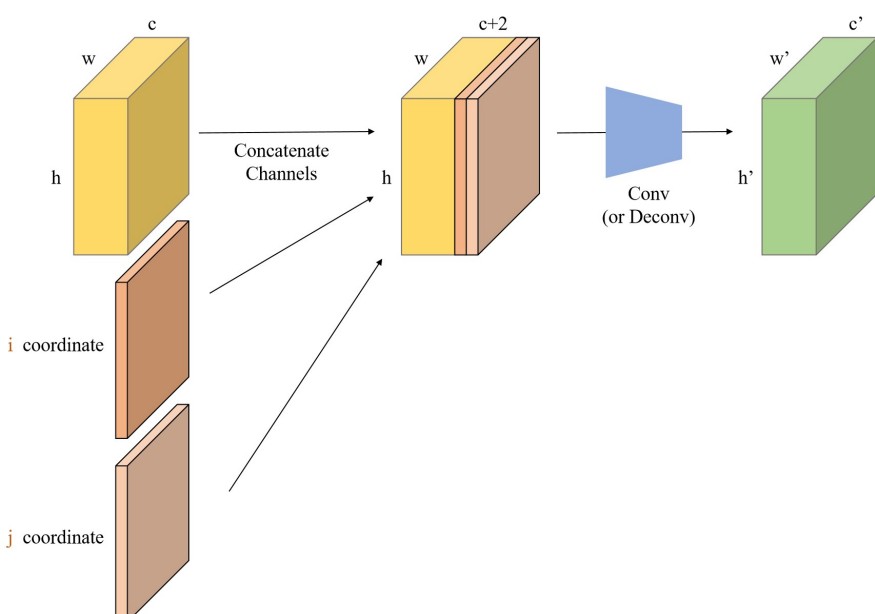

**Figure 5.** The variables $h, w$, and $c$ denote the height, width, and channel dimensions prior to the convolution process, while $h^{'}, w^{'}$, and $c^{'}$ represent the resulting height, width, and channel dimensions after the convolution. The CoordConv technique initially establishes a connection between the supplementary channels and the incoming representation, in order to realize the mapping process. These channels consist of coordinates that are predetermined and cannot be changed. The simplest form of these channels includes one channel for the i-coordinate and another channel for the j-coordinate.

YOLOv5 uses nearest neighbor interpolation for upsampling, which has the advantage of being computationally simple and fast, but ignores the fact that there is rich semantic information in the feature map. In order to better aggregate the pixel domain information and improve the upsampling accuracy, the lightweight CARAFE [64] is introduced to replace the original upsampling method. CARAFE (as shown in Figure 6) is an approach consisting of an upsampling kernel prediction module and a feature reassembly module. Its main goal is to generate reconstructed convolutional kernels to achieve high quality upsampling. The input image size of the method is assumed to be $C \times H \times W$, where C denotes the number of channels and H and W denote the height and width of the image, respectively. In order to reduce the computational complexity, CARAFE first performs channel compression on the input channel C using $1 \times 1$ convolution to obtain $C_m$. Then, in the content encoder module, a convolution operation is performed on the feature maps output from the previous module to generate a recombination kernel $K^2 \times d^2$ of size $H \times W$. The weights of these convolution kernels reflect correlations between different channels. Next is the operation of the content-aware recombination module. First, a weighted summation operation is performed on the recombination kernels, to recombine the features within a local region. Then, a region of size $k \times k$ is selected for convolution for each pixel of the original input feature map. Finally, the convolution result is subjected to an inner product operation with the recombination kernel, to obtain an output feature map of size $C \times \delta H \times \delta W$, where $\delta$ denotes the upsampling coefficient. To make the downsampling operation more focused on positional information, we replace the convolution calculation in CARAFE with CoordConv convolution. The structure of the improved YOLOv5 is shown in Figure 7.

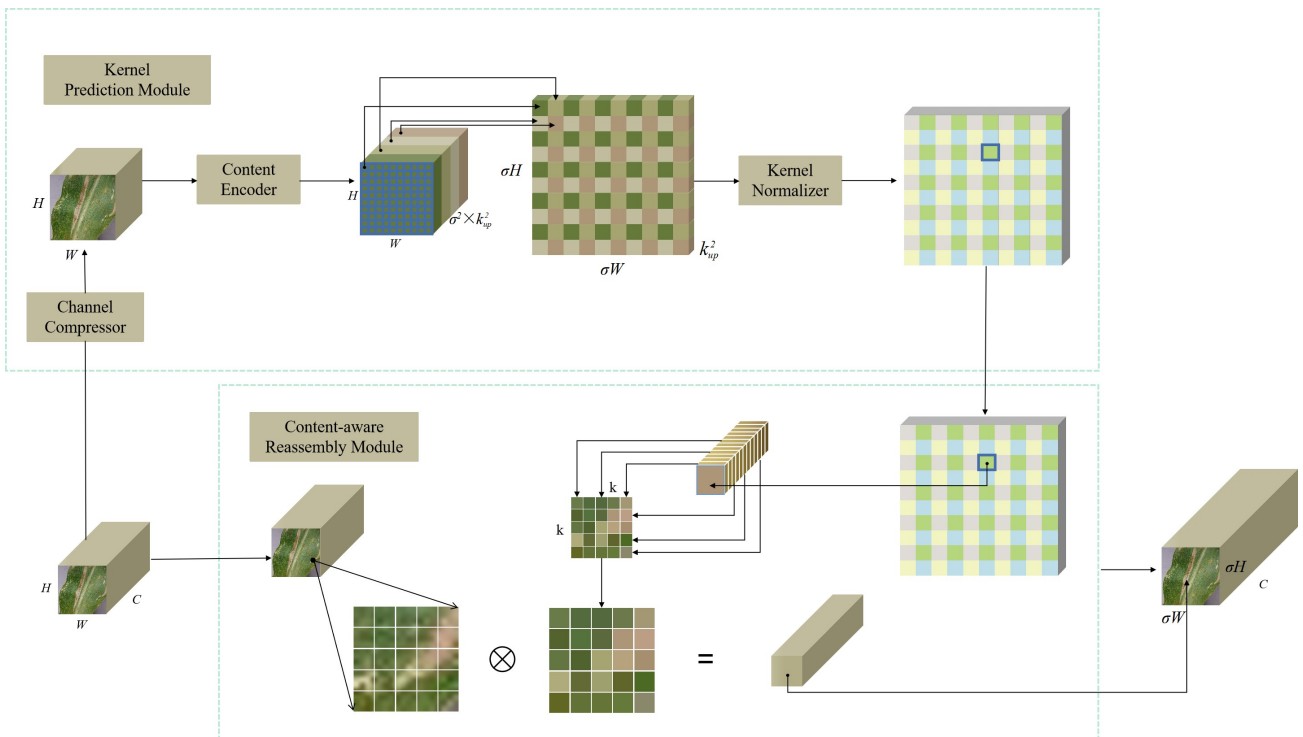

**Figure 6.** CARAFE structure.

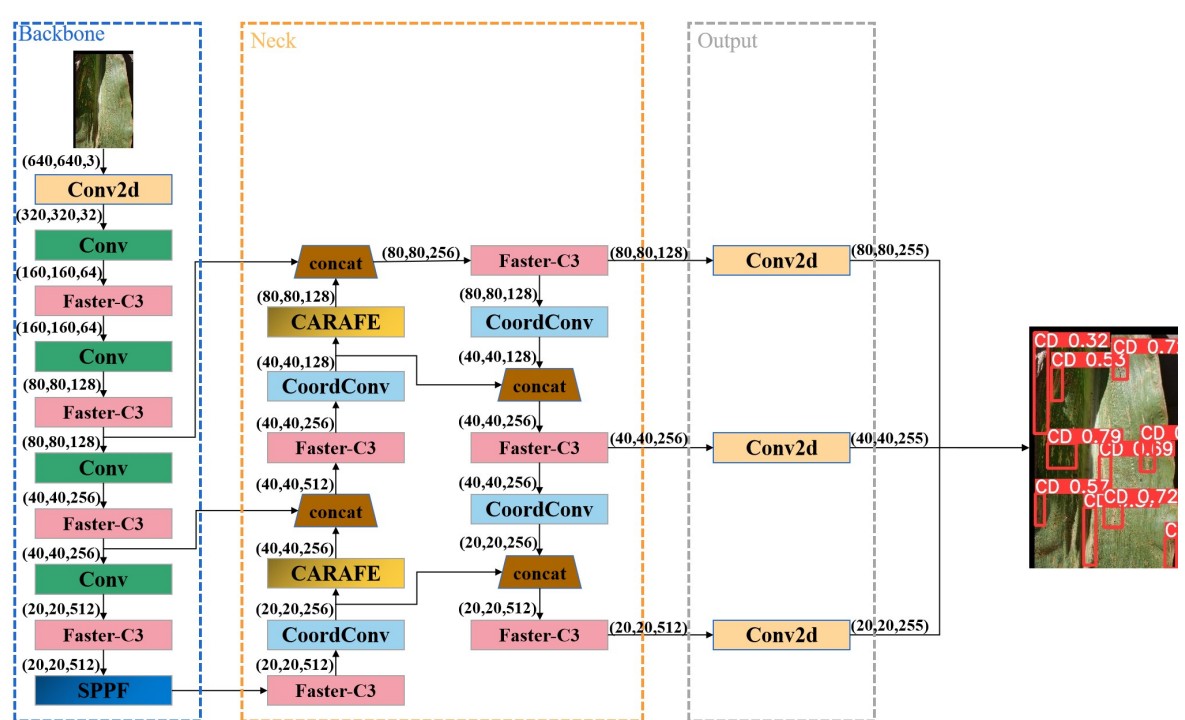

**Figure 7.** Improved YOLOv5 network structure.

## 2.4. Channel-Wise Knowledge Distillation

Knowledge distillation is an effective way to improve the detection accuracy of lightweight detection algorithms. Existing knowledge distillation methods typically use point-by-point alignment, or the alignment of structured information between spatial locations, but the channel contains a large amount of knowledge that is overlooked during the distillation process. Channel-wise knowledge distillation [65] allows better utilization of the knowledge in each channel, and the activation of the corresponding channel between

the teacher and student networks should be adjusted in small steps. To do this, the activation of the channels is first transformed into a probability distribution, so that we can use probabilistic distance measures, such as KL scattering, to measure the differences. Denote the teacher and student networks as $T$ and $S$, and the activation mappings for $T$ and $S$ as $y^T$ and $y^S$, respectively. Channel-wise distillation losses can be expressed in a general form as

$$\varphi(\phi(y^T), \phi(y^S)) = \varphi(\phi(y_c^T), \phi(y_c^S)). \tag{3}$$

Use $\phi(.)$ to convert the activation values into a probability distribution, as expressed in Equation (4):

$$\phi(y_c) = \frac{exp(\frac{y_{c,i}}{\tau})}{\sum_{i=1}^{W \times H} exp(\frac{y_{c,i}}{\tau})}. \tag{4}$$

where $c = 1, 2, \ldots, C$ indexes the channel; $i$ indexes the spatial location of the channel. $\tau$ is a hyperparameter (temperature). If we use a larger $\tau$, the probability becomes softer, which means we focus on a wider spatial region for each channel. By applying softmax normalization, we remove the effect of the magnitude scale between large and compact networks. If there is a mismatch in the number of channels between the teacher and student, a $1 \times 1$ convolutional layer is used to upsample the number of channels in the student network. In this study, the lightweight maize disease detection algorithm proposed through YOLOv5m distillation was used. Specifically, the three feature layers extracted from the backbone network (small, medium, and large) were subjected to knowledge distillation. In this study, a light maize disease detection algorithm utilizing the YOLOv5m distillation method was used. Specifically, knowledge was refined from three feature layers (small, medium and large) extracted from the backbone network, as shown in Figure 8.

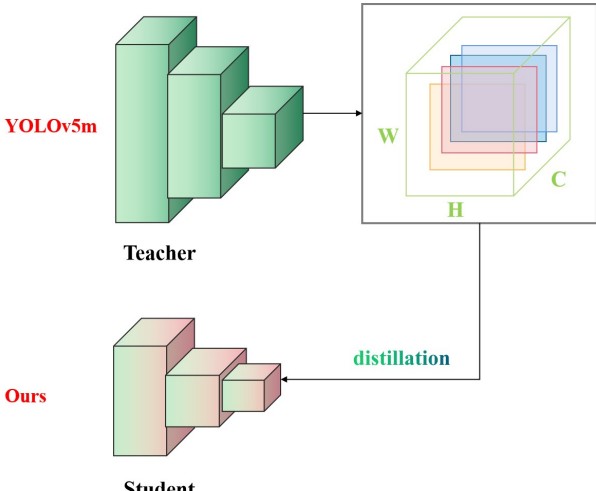

**Figure 8.** Channel-wise knowledge distillation method (this study uses the lightweight detection algorithm with YOLOv5m distillation).

By integrating these enhanced techniques, we present a novel algorithm for the detection of maize leaf diseases, which is characterized by its lightweight nature. The algorithm enhances the precision of detection, while simultaneously decreasing the quantity of model parameters and FLOPS. The pseudocode for this algorithm is presented in Algorithm 1.



---

**Algorithm 1** Pseudocode for the improved YOLOv5 algorithm

---

**Input:** maize leaf images Img_maize, Bounding box coordinates $B_x$, $B_y$, width $B_w$, height $B_h$, trained YOLOv5m weights

**Output:** Class probabilities $P_c$ and Predicted Bounding box coordinates

1:  **Initialize:** Img_robot = Img_train(70%) + Img_test(20%) + Img_validation(10%);
2:  batch_size = 16;
3:  epochs E = 300;
4:  **Training Stage:**
5:  **for** $i = 1 : E$ **do**
6:      Training on the Img_train with the improved YOLOv5 algorithm (As shown in Figure 7);
7:      Load trained YOLOv5m weights;
8:      Calculation of the loss function by the knowledge distillation (Section 2.4) method.
9:      Evaluating algorithm using Img_validation;
10: **end for**
11: save optimal checkpoint bestweight.pt;
12: **Testing Stage:**
13: **for** $testImage : Img\_test$ **do**
14:     predict y = f(Pc, Bw, Bh, Bx, By)
15:     No of predictions = $13 \times 13 \times (5 \times 3 + c)$
16:     Display class c and Pc
17: **end for**

---

*2.5. Experimental Settings*

2.5.1. Experimental Platform

The experimental environment for this study was as follows: an AMD Ryzen Threadripper PRO 5955WX 16-Cores processor and an NVIDIA RTX A5500 The whole machine is produced from Lenovo Group, Beijing, China. graphics card, with 64 GB of RAM and 24 GB of video memory, respectively. Operating system: Windows 10. Deep learning framework: pytorch 1.7.1 (cuda 11.3). Language: python 3.8.1. Dataset annotation tool: LabelImg 1.8.0.

The edge computing device used in this study was Manifold 2-g manufactured by DJI with an NVIDIA Jetson TX2 processor,the whole machine is manufactured from Shenzhen DJI Innovation Technology Co Ltd, Shenzhen, China. Ubuntu 18.04 system, and 8 g of RAM.

2.5.2. Evaluating Indicators

The accuracy metrics included *Precision*, *Recall*, and *mAP* (mean average precision), which are defined as follows:

$$Precision = \frac{TP}{TP + FP}, \tag{5}$$

$$Recall = \frac{TP}{TP + FN}. \tag{6}$$

In Equations (5) and (6), the term true positive ($TP$) represents a detection result that is accurate. The predicted bounding box is considered a true positive ($TP$) when it closely matches the annotated bounding box and when the intersection over union (IoU) value between them exceeds a specified threshold for intersection ratio. On the other hand, a false positive ($FP$) occurs when the predicted bounding box, despite being in close proximity to the annotated bounding box, has an IoU value below the specified threshold. A false negative ($FN$) refers to a situation where the detection network fails to produce a predicted bounding box that is in close proximity to the labeled bounding box.

Given a dataset with $N$ samples, out of which $m$ samples are positive cases, we can calculate $m$ recall values by dividing the number of correctly detected positives by the total number of positives for different recall thresholds. These *Recall* values range from $1/m$ to 1, with increments of $1/m$. For each *Recall* value R, we can determine the corresponding

maximum precision $P$. The maximum precision is the highest precision achievable while maintaining a recall value greater than or equal to R. By averaging these m precision values, we obtain the average precision ($AP$) value. $AP$ is a measure of how well the model performs in terms of precision across different recall levels. Equation (7) can be used to determine the $AP$ value and is typically used as a metric to evaluate the performance of object detection algorithms.

$$AP = \frac{1}{m} \sum_{i}^{m} P_i = \frac{1}{m} \times P_1 + \frac{1}{m} \times P_2 + \cdots + \frac{1}{m} \times P_m. \tag{7}$$

$mAP$ is calculated by averaging the $AP$ of all classes in the dataset. The following is the Equation (8) for $mAP$:

$$mAP = \frac{1}{C} \sum_{C}^{j} P_j. \tag{8}$$

In addition to object detection, the task also includes evaluating the accuracy of boundary regression. The $IoU$ is calculated as in Equation (9).

$$IoU = \frac{S_1}{S_2}. \tag{9}$$

In object detection, $S_1$ represents the overlapping area between the predicted box and the actual box, while $S_2$ represents the total area occupied by both boxes. The metric $mAP(0.5)$ measures the mean average precision, with an $IoU$ threshold set to 0.5, while $mAP(0.5:0.95)$ calculates the average $mAP$ across different $IoU$ thresholds ranging from 0.5 to 0.95, in increments of 0.05.

To evaluate the algorithm's performance on hardware, this study considered several metrics, including the average processing time for detection, frames per second (FPS), the number of model parameters, and floating point operations (FLOPs). These metrics provide insights into the algorithm's computational efficiency and resource requirements.

### 2.5.3. Hyperparameter Setting

The experiments were conducted using stochastic gradient descent (SGD), with a batch-size set to 16, epoch set to 300, input image size of 640*640, learning rate set to 0.001, weight decay factor set to 0.0005, and momentum factor set to 0.937. Mosaic data enhancement, Mixup [66], and CosineAnnealing [67] methods were used.

## 3. Results and Discussion

### 3.1. Ablation Experiment Results

Table 3 shows the results of the ablation experiments. Each individual enhancement was incorporated independently into the foundational YOLOv5s model. When exclusively utilizing the enhanced Faster-C3 model, there was a reduction in precision, recall, mAP(0.5), and mAP(0.5:0.95) of 0.5%, 0.7%, 0.3%, and 1.6%, respectively, in comparison to the YOLOv5s model. However, there was a decrease in the number of parameters by 17.5% and FLOPs by 3.1G. The enhancement of solely the neck network resulted in an increase of 1.1% in recall, as well as 0.5% and 0.4% in mAP(0.5) and mAP(0.5:0.95), respectively, when compared to YOLOv5s. Notably, the number of parameters and FLOPs experienced minimal alteration. When solely employing knowledge distillation, there was no alteration in the quantity of model parameters and FLOPs. However, there was an improvement in precision, recall, mAP(0.5), and mAP(0.5:0.95) by 2.6%, 3.0%, 2.5%, and 2.2%, correspondingly. The cumulative enhancements yielded a 2.4% increase in precision, a 2.7% increase in recall, a 3.8% increase in mAP(0.5), and a 1.5% increase in mAP(0.5:0.95). Additionally, there was a reduction of 15.5% in the number of model parameters and a decrease of 2.9G in FLOPs when compared to YOLOv5s.

The observed outcomes indicate that by solely changing the Faster-C3 module, there was a noteworthy drop in the number of model parameters and FLOPs, with only a

minor decline in accuracy. This suggests that the proposed modification successfully reduces the overall complexity of the model. The observed substantial enhancement in recall, specifically when focusing on enhancing the neck network exclusively, provides compelling evidence that this modification effectively enhances the network's ability to recognize and understand spatial information, while still preserving a strong resemblance to the original model's size. Exclusive utilization of the knowledge distillation technique has the potential to enhance the overall accuracy of the model, while simultaneously maintaining a constant number of model parameters and floating point operations (FLOPs). Following the integration of many enhancement strategies, the assessment metrics of YOLOv5s experienced complete improvements, yielding good outcomes. In summary, the efficacy of the enhancement technique described in this study has been demonstrated.

**Table 3.** Results of ablation experiments. "+Faster-C3" indicates improved CSP; "+KD" indicates the use of a knowledge distillation algorithm; "+Neck" indicates an improved neck network.

| +Faster-C3 | +Neck | +KD | Precision | Recall | mAP(0.5) | mAP(0.5:0.95) | Parameters | FLOPs(G) |
|:---:|:---:|:---:|:---:|:---:|:---:|:---:|:---:|:---:|
| | | | 72.9% | 66.3% | 71.0% | 33.1% | 7,023,610 | 15.8 |
| ✓ | | | 72.4% | 65.6% | 70.7% | 31.5% | 5,793,082 | 12.7 |
| | ✓ | | 72.9% | 67.4% | 71.5% | 33.5% | 7,105,606 | 15.9 |
| | | ✓ | 75.5% | 69.3% | 73.5% | 35.3% | 7,023,610 | 15.8 |
| ✓ | ✓ | ✓ | 75.3% | 69.0% | 74.8% | 34.6% | 5,933,954 | 12.9 |

### 3.2. Comparative Results of Different Knowledge Distillation Methods

The channel-wise knowledge distillation (cwd) method was employed in this study for the purpose of distillation. In order to conduct a comparative analysis of the efficacy of various knowledge distillation techniques, the YOLOv5s model was subjected to distillation using three distinct methods: mgd [68], mimic [69], and cwd. These methods were applied within a consistent experimental framework. The experimental results are depicted in Figure 9. The figure illustrates that the other knowledge distillation methods, namely mimic and mgd, exhibited a comparable training accuracy. However, the cwd method demonstrated a superior training accuracy and exhibited faster improvement.

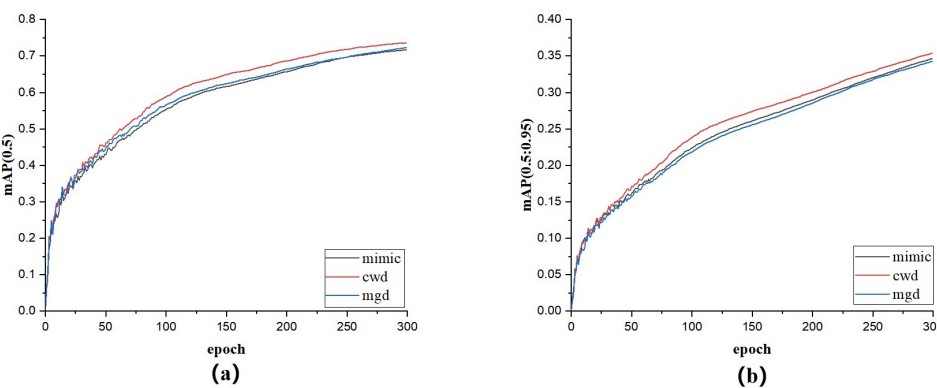

**Figure 9.** Accuracy of the different knowledge distillation methods. (**a**) Training 300 epoch of mAP(0.5). (**b**) Training 300 epoch of mAP(0.5,0.95).

While both the mgd and mimic methods have demonstrated efficacy in enhancing accuracy, their effectiveness mostly hinges on the utilization of soft labels derived from model outputs or the decisions made by the teacher model. However, it is worth noting that these approaches may not fully capture the nuanced information inherent in the teacher model. CWD places a focus on the effective transfer of knowledge across different channels. This transfer is accomplished through a systematic adjustment of the activation process, which occurs incrementally. This facilitates the student model in effectively capturing significant features and information from the instructor model across all channels.

The process of knowledge distillation is facilitated by channel-wise distillation (CWD), which involves the modification of channel activations between the teacher and student networks, in order to optimize the utilization of knowledge inside each channel. Therefore, the utilization of cwd distillation greatly enhances the precision of the algorithm and expedites the learning capacity of the model.

### 3.3. Comparison of Different Versions of YOLOv5

This research paper presents an enhanced and optimized lightweight detection algorithm utilizing the YOLOv5s framework. To assess the efficacy of the proposed algorithm across multiple iterations of YOLOv5, a series of experiments were conducted independently for each version of YOLOv5, employing an identical experimental configuration. The experimental results for various versions of YOLOv5 are presented in Table 4. Although the proposed algorithm exhibits a higher number of parameters and FLOPs, YOLOv5n demonstrates significantly lower values in these aspects. Nevertheless, the algorithm under consideration exhibited an increase of 5.1%, 12.5%, 16.9%, and 11.6% in terms of precision, recall, mAP(0.5), and mAP(0.5:0.95), respectively, when compared to the YOLOv5n model. The proposed algorithm has a significantly lower number of model parameters compared to YOLOv5m and YOLOv5l, with the former being 3.5-times higher and the latter being 7.8-times higher. The mean average precision (mAP) at the intersection over union (IoU) threshold of 0.5 and the mAP across the IoU range of 0.5 to 0.95 exhibited improvements of approximately 3% and 10% over the proposed algorithm, respectively.

In addition to the aforementioned findings, it is worth noting that while YOLOv5m and YOLOv5l exhibit commendable detection accuracy, their practical applicability is limited due to the substantial equipment demands resulting from the significant escalation in both the parameter count and FLOPs. The YOLOv5n model demonstrated a notable efficiency in terms of model parameters and FLOPs, yet this compromises a substantial portion of its detection accuracy. Although YOLOv5m and YOLOv5l were about 5% higher compared to our proposed algorithm in the four metrics of model precision, recall, mAP(0.5), and mAP(0.5:0.95), the number of parameters was 3.5- and 7.8-times higher than the proposed algorithm, and the FLOPS were 3.7- and 8.3-times higher than the proposed algorithm, respectively. Despite the fact that our proposed algorithms exhibited an accuracy metric lower by around 5%, this improvement came at the expense of a significantly larger performance measure. It is evident that our algorithms offer a more favorable balance between accuracy and performance.

**Table 4.** Comparison of different versions of YOLOv5.

| Model | Precision | Recall | mAP(0.5) | mAP(0.5:0.95) | Parameters | FLOPs(G) |
|---|---|---|---|---|---|---|
| YOLOv5n | 60.2% | 56.5% | 57.9% | 23.0% | 1,769,802 | 4.2 |
| YOLOv5s | 73.8% | 65.6% | 71.1% | 32.8% | 7,023,610 | 15.8 |
| YOLOv5m | 79.7% | 59.9% | 76.6% | 42.2% | 20,869,098 | 47.9 |
| YOLOv5l | 80.7% | 68.6% | 76.5% | 44.9% | 46,129,818 | 107.7 |
| Ours | 75.3% | 69.0% | 74.8% | 34.6% | 5,933,954 | 12.9 |

### 3.4. Statistical Significance Test (t-Test)

In this section, statistical significance tests (specifically *t*-tests) are employed to evaluate the presence of a statistically significant disparity in the performance metrics of the proposed model. The *t*-test is a widely employed statistical hypothesis test that is utilized to evaluate the presence of a statistically significant difference between two groups or samples. This is accomplished through the computation of the *t*-statistic and its associated *p*-value. The *t*-statistic is a statistical metric that quantifies the disparity between the means of two groups, while considering the variability within each group. This metric represents the magnitude of the standard error in the difference between the means of the two groups. The *p*-value is a statistical measure that represents the likelihood of observing a result as

extreme as the *t*-statistic we achieved, assuming that the null hypothesis is true. The null hypothesis posits that there exists no statistically significant difference between the means of the two sets. In statistical analysis, it is customary to employ a significance level of 0.05 (or 5%) as the critical value for establishing statistical significance. When the *p*-value is smaller than 0.05, it is appropriate to reject the null hypothesis and infer that there exists a statistically significant disparity between the means of the two sets.

This study utilized a paired *t*-test to assess the efficacy of the proposed methodology over multiple iterations of YOLOv5. The assessment was performed utilizing two metrics: mean average precision (mAP) at a threshold of 0.5, and the parameter count. As the performance of the model becomes closer, the absolute value of the *t*-statistic decreases. Based on the results displayed in Table 5, our model's *t*-statistic demonstrates a positive value and a much greater magnitude compared to YOLOv5n and the other models. Furthermore, the obtained *p*-value (0.02268776414333465) falls below the commonly accepted threshold of 0.05. Therefore, it can be deduced that a statistically significant discrepancy existed in the means of the two groups or samples. The performance of the suggested model, as indicated by its mean average precision (mAP) metrics at a threshold of 0.5, exhibited a much higher level of effectiveness in comparison to YOLOv5n. The comparative study entailed the assessment of the suggested model in comparison to YOLOv5s, YOLOv5m, and YOLOv5l. The *t*-statistic provided evidence that the proposed model demonstrates the greatest resemblance to YOLOv5m in relation to mean average accuracy (mAP) when the threshold was set at 0.5. The *p*-values for YOLOv5s, YOLOv5m, and YOLOv5l were all above 5%. This implies that the proposed model did not demonstrate a statistically significant distinction from YOLOv5s, YOLOv5m, and YOLOv5l in relation to the accuracy metric mAP(0.5). As indicated in Table 6, the *t*-statistic values reveal that the parameter count of the proposed model closely approximates that of YOLOv5s, being smaller in magnitude. Additionally, the parameter count of the proposed model surpassed that of YOLOv5n, albeit being significantly smaller than both YOLOv5m and YOLOv5l. Based on the observation that all *p*-values were less than 0.05, it is evident that a statistically significant distinction between the suggested model and the alternative models exists, in terms of the parameter representing the number of indicators.

In conjunction with the aforementioned findings, the suggested model exhibited no substantial disparity in the accuracy metric mAP(0.5) when compared to YOLOv5s, YOLOv5m, and YOLOv5l. In contrast, the suggested model exhibited a notable disparity in the performance index parameters when compared to YOLOv5s, YOLOv5m, and YOLOv5l. The results demonstrate that our suggested model achieved a comparable accuracy to the larger models, while maintaining a significantly reduced model volume. This indicates a favorable trade-off between detection accuracy and computing efficiency.

**Table 5.** The results of the statistical significance tests were used to compare the proposed model with other variants of YOLOv5 for mAP metrics in terms of *t*-statistics and *p*-values.

| Model | *t*-Statistic | *p*-Value |
|---|---|---|
| oursVSYOLOv5n | 6.525515982455876 | 0.02268776414333465 |
| oursVSYOLOv5s | 1.9166296949998196 | 0.19533732882126978 |
| oursVSYOLOv5m | −1.278724026182012 | 0.32931816306533834 |
| oursVSYOLOv5l | −1.4263994477758972 | 0.28986635074139566 |

**Table 6.** The results of the statistical significance tests were used to compare the proposed model with other variants of YOLOv5 for parameters metrics in terms of *t*-statistics and *p*-values.

| Model | *t*-Statistic | *p*-Value |
|---|---|---|
| oursVSYOLOv5n | 198.92926732280444 | $2.526889043803266 \times 10^{-5}$ |
| oursVSYOLOv5s | −53.54869293825585 | 0.00034855806759837796 |
| oursVSYOLOv5m | −677.3082614762818 | $2.1798461056586686 \times 10^{-6}$ |
| oursVSYOLOv5l | −1578.6111663118097 | $4.012817397184073 \times 10^{-7}$ |

### 3.5. Comparison Results of Different Lightweight Detection Models

In order to establish the efficacy of our proposed methodology, a comparative analysis was conducted between our approach and the existing light detection algorithms, using identical experimental conditions. The empirical findings are presented in Table 7. At a mean average precision (mAP) threshold of 0.5, our proposed algorithm demonstrated superior performance compared to YOLOv3-tiny, YOLOv4-tiny, YOLOv5-lite-g, and YOLOv5-lite-e, with improvements of 18.2%, 17.6%, 25.1%, and 51.4%, respectively. In terms of mean average precision (mAP) with a range of 0.5 to 0.95, our proposed algorithm demonstrated superior performance compared to YOLOv3-tiny, YOLOv4-tiny, YOLOv5-lite-g, and YOLOv5-lite-e, with improvements of 11.3%, 10.1%, 15.9%, and 27.6%, respectively. YOLOv3-tiny, YOLOv4-tiny, and YOLOv5-lite-g exhibited similar parameter counts and FLOPs as our proposed algorithm, while YOLOv5-lite-e demonstrated significantly lower parameter counts and FLOPs in comparison to our proposed algorithm.

To assess the efficacy of our suggested approach, we performed a comparative analysis between our method and contemporary lightweight detection algorithms, employing identical experimental conditions. The experimental findings are presented in Table 7. At a threshold of 0.5 for mean average precision (mAP), our proposed approach demonstrated an improved performance in comparison to YOLOv3-tiny, YOLOv4-tiny, YOLOv5-lite-g, and YOLOv5-lite-e. The observed improvements were 18.2%, 17.6%, 25.1%, and 51.4%, correspondingly. Within the range of 0.5 to 0.95 mean average precision (mAP), our proposed approach demonstrated superior performance in comparison to YOLOv3-tiny, YOLOv4-tiny, YOLOv5-lite-g, and YOLOv5-lite-e. Specifically, our technique exhibited improvements of 11.3%, 10.1%, 15.9%, and 27.6% over these respective models. The YOLOv3-tiny, YOLOv4-tiny, and YOLOv5-lite-g models exhibited comparable parameter numbers and FLOPs to our proposed algorithm. Conversely, the YOLOv5-lite-e model demonstrated notably lower parameter numbers and FLOPs in comparison to our proposed approach. In comparison to YOLOv7-tiny and YOLOv8n, the suggested technique exhibited similarities in terms of the parameter count and FLOPs. However, it demonstrated worse performance in metrics such as the precision, recall, mAP(0.5), and mAP(0.5:0.95). The results demonstrated that our suggested model had an exceptional detection accuracy, compared to models with similar parameter numbers and FLOPs. The number of parameters and FLOPs in YOLOv8s is approximately twice as high as that of the proposed model. While YOLOv8s exhibited a 4.4% higher mean average precision (mAP) at the intersection over union (IoU) threshold of 0.5 to 0.95, and it demonstrated a 3.7%, 5.6%, and 5.2% lower precision, recall, and mAP(0.5), respectively. This observation shows that the suggested model achieved a more optimal trade-off between accuracy and recall in the task of target identification, while also attaining a higher average accuracy when evaluated at an intersection over union (IoU) threshold of 0.5. While YOLOv8s demonstrated a satisfactory performance within a broader range of IoU thresholds (0.5 to 0.95), its performance was somewhat subpar when subjected to stricter criteria.

In order to illustrate the applicability of our suggested method, we conducted a comparative analysis of many lightweight detection algorithms using the publicly available PascalVOC dataset. The outcomes of our experiments are presented in Table 8. In comparison to YOLOv5m, the suggested model exhibited a higher precision, recall, mAP(0.5), and mAP(0.5:0.95) by 6.2%, 1.6%, 3.8%, and 6.9%, respectively. However, it is important to note that the proposed model had a much larger number of model parameters and FLOPs; approximately 3.5- and 3.7-times higher, respectively, compared to the proposed model. The results suggest that YOLOv5m demonstrated enhancements in precision, recall, mAP(0.5), and mAP(0.5:0.95) measures compared to the proposed model. However, these improvements were accompanied by a more intricate model architecture and an increased number of parameters and FLOPs, which should be taken into consideration. These findings indicate that the suggested model achieved a favorable trade-off between model complexity and performance in comparison to YOLOv5s and YOLOv7-tiny. Additionally, the proposed model exhibited a minor advantage in terms of performance measures. The

YOLOv8s model exhibited approximately a 2.5-times greater parameter count and FLOPs compared to the suggested model. However, the proposed model demonstrated improved efficiency in terms of accuracy, recall, mean average accuracy (mAP) at IoU threshold 0.5, and mAP across the range of IoU thresholds from 0.5 to 0.95. Specifically, the proposed model achieved improvements of 1.4% in precision, 2.1% in recall, 1.7% in mAP(0.5), and 6.1% in mAP(0.5:0.95). The suggested model exhibited a notable superiority over YOLOv8s in relation to the quantity of parameters and FLOPs. However, the enhancements observed in the performance metrics of precision, recall, mAP(0.5), and mAP(0.5:0.95) were not statistically significant. Significantly, the performance of the suggested model on the PascalVOC public dataset demonstrated a high level of consistency with the maize leaf disease dataset, hence providing evidence of the generalizability of the proposed strategy.

Figure 10a,b shows the single image inference time and FPS of the relevant lightweight detection algorithms on edge computing devices. To give a fair indication of the time performance of the detection algorithms, none of the tested detection algorithms used any detection acceleration techniques. From Figure 10a, it can be seen that YOLOv5m took the longest time to reason about a single image, YOLOv8s was the second longest, and our proposed algorithm took the shortest time. This indicates that the proposed algorithm had a faster inference on edge computing devices. From Figure 10b, our proposed algorithms, YOLOv4-tiny, YOLOv5m, and YOLOv8s achieved 5.03FPS, 4.18FPS, 2.09FPS, and 3.729FPS, respectively on edge computing devices. It can be seen that the proposed algorithms achieved the highest frame rate, i.e., the highest processing speed, on this device. In summary, the proposed algorithm shows a shorter inference time and higher frame rate on edge computing devices as compared to the comparative lightweight algorithms. This means that it may be more suitable for edge computing scenarios in practical applications.

Based on the aforementioned findings, our suggested methodology demonstrated superior detection accuracy in comparison to the existing lightweight detection methods, while simultaneously preserving a reduced number of model parameters and FLOPs. Hence, by evaluating the algorithm's dependability, robustness, and adaptability, it becomes evident that our method possesses the attribute of being lightweight, without compromising its excellent performance. The method demonstrated superior performance compared to the existing lightweight detection algorithms, as seen by its improved detection accuracy and detection performance metrics.

**Table 7.** Comparison results of the different lightweight detection models.

| Model | Precision | Recall | mAP(0.5) | mAP(0.5:0.95) | Parameters | FLOPs(G) |
|---|---|---|---|---|---|---|
| YOLOv3-tiny | 62.2% | 54.4% | 56.6% | 23.3% | 8,675,932 | 12.9 |
| YOLOv4-tiny | 63.4% | 55.8% | 57.2% | 24.5% | 5,883,356 | 16.2 |
| YOLOv5-lite-g | 42.4% | 43.3% | 38.4% | 12.6% | 7,023,610 | 15.8 |
| YOLOv5-lite-e | 31.1% | 34.6% | 23.4% | 7% | 1,649,853 | 4.0 |
| YOLOv7-tiny | 59.0% | 54.8% | 56.9% | 22.9% | 6,018,420 | 13.1 |
| YOLOv8n | 59.9% | 51.4% | 64.9% | 25.6% | 3,006,623 | 8.1 |
| YOLOv8s | 71.8% | 63.4% | 69.6% | 39.0% | 11,127,519 | 28.4 |
| Ours | 75.3% | 69.0% | 74.8% | 34.6% | 5,933,954 | 12.9 |

**Table 8.** Comparison results of the different lightweight detection models on the PascalVOC dataset.

| Model | Precision | Recall | mAP(0.5) | mAP(0.5:0.95) | Parameters | FLOPs(G) |
|---|---|---|---|---|---|---|
| YOLOv5s | 80.2% | 68.8% | 75.9% | 51.4% | 7,023,610 | 15.9 |
| YOLOv5m | 84.3% | 72.6% | 80.6% | 59.7% | 20,929,713 | 48.1 |
| YOLOv7-tiny | 76.8% | 69.3% | 75.5% | 50.5% | 6,059,010 | 13.2 |
| YOLOv8s | 83.5% | 73.1% | 81.0% | 62.7% | 11,127,519 | 28.5 |
| Ours | 82.1% | 71.0% | 79.3% | 56.6% | 5,933,954 | 12.9 |

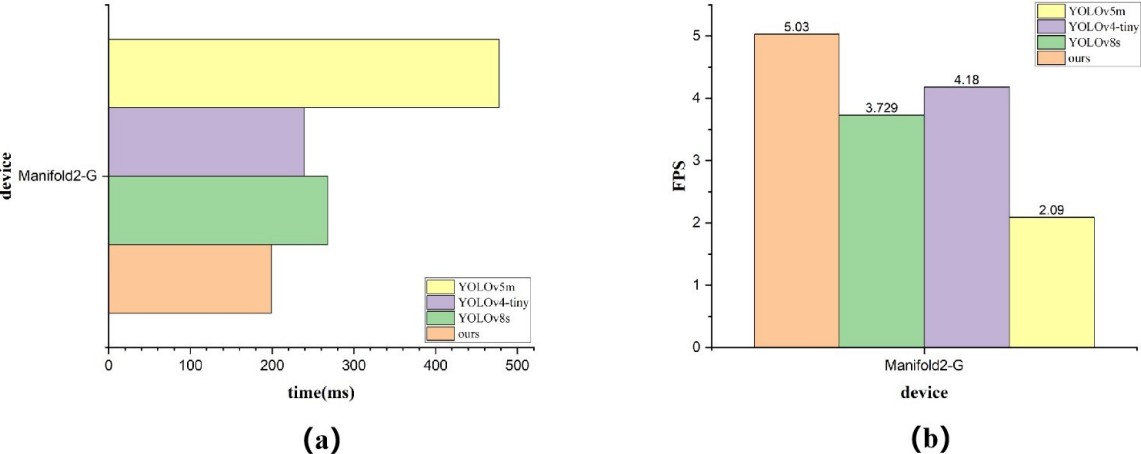

**Figure 10.** Comparison of the temporal performance of the different models. (**a**) represents the time for the algorithm to reason about a single image on the edge device and (**b**) represents the FPS of the algorithm on the edge device.

## 4. Conclusions

In order to tackle the challenges associated with the detection and identification of maize leaf diseases in practical settings, we presented a novel algorithm for detecting light maize leaf diseases. This algorithm is built upon an enhanced version of the YOLOv5 model. Initially, the original CSP module was substituted with the Faster-C3 module, which achieved a reduction in the quantity of convolution operations. This reduction effectively decreased the number of model parameters and FLOPs.Additionally, we incorporated CoordConv within the neck network to enhance the capturing of positional information. Furthermore, we proposed an enhanced CARAFE upsampling operation to extract a broader spectrum of semantic information, thereby enhancing the precision of maize leaf disease detection. Finally, the suggested algorithm for detecting lightweight maize leaf disease was refined through the implementation of the channeled knowledge distillation technique, while maintaining the original model size, in order to enhance the algorithm's accuracy. The experimental findings demonstrated that our proposed algorithm exhibited a reduction of 2.9G FLOPs and had 15.5% fewer model parameters in comparison to the conventional YOLOv5s. Additionally, the mAP(0.5) and mAP(0.5:0.95) metrics displayed enhancements of 3.8% and 1.5%, respectively. These results indicated that the enhanced algorithm effectively enhanced performance. The algorithm we developed exhibited a favorable equilibrium between detection accuracy and computational efficiency when compared to other lightweight algorithms. This finding underscores the algorithm's strong generality.

Although our proposed method has advantages over other lightweight detection algorithms on resource-limited edge devices, it still has great potential. The method does not rely on any accelerated detection techniques to be implemented on these devices. Another key issue is how to ensure the robustness of the model in real-world environments, including the effects of factors such as weather conditions and light changes. The next step in our research will focus on incorporating agricultural robots to recognize maize leaf disease detection. In addition, an intriguing study direction is exploring the use of semi-supervised learning techniques to enhance the algorithms, to ensure accurate detection results across a wide range of weather conditions in real-world environments.

**Author Contributions:** Conceptualization, J.G. and Y.H.; methodology, Z.C. and J.L.; software, Z.C. and Y.H.; validation, J.G., J.L. and Y.H.; formal analysis, G.L. and Y.H.; investigation, J.G. and G.L.; resources, Y.H. and G.L.; data curation, J.L. and Y.H.; writing—original draft preparation, Y.H. and J.L.; writing—review and editing, Y.H., Z.C. and J.L.; visualization, Y.H. and J.L.; supervision, J.G. and G.L.; project administration, J.G.; funding acquisition, J.G. All authors have read and agreed to the published version of the manuscript.

**Funding:** This study was funded by the Scientific Research Project of Jilin Provincial Education Department (JJKH20230764KJ).

**Institutional Review Board Statement:** Not applicable.

**Informed Consent Statement:** Not applicable.

**Data Availability Statement:** No new data were created or analyzed in this study. Data sharing is not applicable to this article.

**Conflicts of Interest:** The authors declare no conflict of interest.

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
