# Peer review of "Lightweight One-Stage Maize Leaf Disease Detection Model with Knowledge Distillation"

_agriculture, doi:10.3390/agriculture13091664_

Round 1
Reviewer 1 Report
see the attachment

Author Response
Dear Editor,
We appreciate you and the reviewers for your precious time in reviewing our paper and providing valuable comments. It was your valuable and insightful comments that led to possible improvements in the current version. The authors have carefully considered the comments and tried our best to address every one of them. We hope the manuscript after careful revisions meet your high standards.
The authors welcome further constructive comments if any. Below we provide the point-by-point responses. All modifications in the manuscript are indicated in red for deletions and in blue for additions.
Sincerely,
Yanxin Hu

Reviewer 2 Report
Attached herwith.

Author Response

(The authors gave the same response as above.)

Reviewer 3 Report
The study proposed a modified light YOLOv5s for corn leaf disease detection. The manuscript itself is well-written, however, it has a few fundamental issues.
The study, just like many other recent ones that I reviewed, is a backward study with no true novelty. It takes an obsolete object detection framework, claims it is not “fast” enough, hence a lighter version of the framework needs to be developed, and the study is justified. YOLO series are developed with the goal of achieving real time performance, processing more than 24 frame per second. Anybody that claims YOLO models are too large to be run on embedded devices, needs to provide tangible evidence to support the claim. Then, the study takes proposed modules from prior studies to modify the obsolete framework, claiming improvements are achieved. Using backbone as an example, of course the framework will become lighter when a small backbone is chosen and the framework performance will suffer. There are absolutely no research questions to be answered by doing such a modification. It is OK to use modules proposed by existing studies in the current study, as long as a new state-of-the-art model performance can be achieved, by comparing to existing state-of-the-art models such as YOLOv8 based on a public benchmark dataset such as MS COCO, instead of comparing to even more obsolete models such as YOLOv3 proposed five years ago.
No knowledge gaps are defined in the manuscript. Research questions of the study are unclear. Literature reviews are general and inappropriate as they should focus on the technical aspect of the modules utilized in the study and how existing studies benefited from using those modules.
Very little information regarding the dataset is provided. Section 2.1 is suspiciously and inappropriately short. How is the “smaller portion of data” collected? Why in the Data Availability Statement says “No new data were created or analyzed in this study”?
Object detection does not seem to be the right method for detecting the leaf disease, as the bounding box annotations make no sense when no explanation was given about how a bounding box should be defined. Semantic segmentation would be more appropriate to precisely cover the diseased leaf areas.
Author Response

(The authors gave the same response as above.)

Round 2
Reviewer 1 Report
The author has diligently incorporated all the valuable feedback provided, resulting in a markedly improved manuscript. The revisions have not only addressed the earlier identified areas of concern but have also contributed to elevating the overall quality and coherence of the document. As a result of these commendable efforts, the manuscript now stands in a significantly enhanced state, poised to convey its content more effectively and engage readers more comprehensively.
Minor editing of English language required
Author Response

(The authors gave the same response as above.)

Reviewer 3 Report
Literature review is still insufficient as it did not dive into discussing the technical advantages and disadvantages as well as the utilization of the modules and methods used to modify YOLOv5 in current literature.
I still don’t see where the authors identified knowledge gaps in current literature to justify the study.
Regarding section 2.1, the definition of bounding box, my previous comment was intended for clarifying annotation guideline, which is still missing and confusing based on figure 1. The plants in the figure have disease symptoms all over their surface, why not use one big bounding box to include the whole image? How was the boundary of each bounding box determined?
Author Response

(The authors gave the same response as above.)
